# High and Hyper: Fentanyl Induces Psychomotor Side-Effects in Healthy Pigs

**DOI:** 10.3390/ani13101671

**Published:** 2023-05-17

**Authors:** Nora Digranes, Henning Andreas Haga, Janicke Nordgreen

**Affiliations:** 1Department of Companion Animal Clinical Sciences, Faculty of Veterinary Medicine, Norwegian University of Life Sciences, 1430 Ås, Norway; andreas.haga@nmbu.no; 2Department of Paraclinical Sciences, Faculty of Veterinary Medicine, Norwegian University of Life Sciences, 1430 Ås, Norway; janicke.nordgreen@nmbu.no

**Keywords:** fentanyl, pig, serotonin, ketanserin, behavior, locomotion

## Abstract

**Simple Summary:**

Pigs used in research are often subjected to procedures for which pain relief is crucial. However, the literature indicates that analgesics are under-reported and under-used in experiments involving pigs. Fentanyl is one of the opioids used to provide post-operative pain relief in experimental pigs. Analgesic requirements are often assessed using behavioral indicators such as activity. However, the effects of fentanyl on behavior in pigs are largely unknown. The aims of the current study were to investigate how fentanyl influences behavior in pigs, and if serotonin could be involved in fentanyl-induced behavioral side effects. The results of this study demonstrate that fentanyl can influence activity level, and induce different repetitive behaviors in pigs. Some of the behavioral changes induced by fentanyl seem to involve the serotonergic system. In conclusion fentanyl might influence behavior through mechanisms unrelated to analgesia. This is important for experimental pigs as it could interfere with pain assessment and determination of analgesic requirements.

**Abstract:**

Analgesic effects of fentanyl have been investigated using behavior. The behavioral effects of fentanyl and possible serotonergic influence are largely unknown. We therefore investigated behavioral effects of fentanyl, with or without the serotonin antagonist ketanserin, in pigs. Fourteen mixed-breed pigs, weighing 17–25 kg were included in a randomised blinded prospective, balanced three-group study. Ten pigs received first 5 and then 10 µg/kg of fentanyl intravenously. Ketanserin at 1 mg/kg or saline was given intravenously as a third injection. Four control pigs received three injections of saline. Behavior was video-recorded. The distance moved was automatically measured by commercially available software, and behaviors manually scored in retrospect. Fentanyl inhibited resting and playing, and induced different repetitive behaviors. The mean (SD) distance moved in the control group and fentanyl group was 21.3 (13.0) and 57.8 (20.8) metres respectively (*p* < 0.05 for pairwise comparison). A stiff gait pattern was seen after fentanyl injection for median (range) 4.2 (2.8–5.1) minutes per 10 min, which was reduced to 0 (0–4) s after ketanserin administration. Conclusion: fentanyl-induced motor and behavioral effects, and serotonergic transmission may be involved in some of them. The psychomotor side effects of fentanyl could potentially interfere with post-operative pain evaluation in pigs.

## 1. Introduction

Pigs are being extensively used as laboratory animals, both for pig-focused research and also as models in human medicine [1]. This applies in particular to translational research where pigs serve as surgical models for training purposes [2,3]. The EU Directive on the protection of animals used for scientific purposes [4] states that animals used for research shall receive anesthesia and analgesia when necessary to minimize any pain, discomfort, fear or stress. However, guidelines for the recognition and treatment of post-operative pain are limited, as is knowledge about species-appropriate analgesia and anesthesia [5]. This is reflected in the literature where only 10% of the reviewed publications reported the use of pain assessment, and only 37% of the papers described post-operative administration of analgesics in pigs [5]. This could be due to under-reporting or under-use. The Animal Research Reporting In Vivo Experiments (ARRIVE) guidelines were published in 2010, aimed at improving the quality of the information included in scientific papers to better be able to assess reliability and reproducibility [6]. However, in spite of these guidelines, the reporting of analgesia in animal models is still inadequate [7]. In a recent publication assessing pre-clinical studies of osteochondral lesions using large animal models, only 1% of the reviewed literature provided enough information on analgesia and anesthesia for the procedures to be repeatable [8].

Pain assessment and treatment in pigs kept for meat production have been studied and described to some extent [9,10], and the results could be applicable to pigs used as surgical models. However, these investigations focus primarily on procedures carried out on a large scale at commercial farms such as surgical castration, and the analgesics tested are limited to those that are marketed for food-producing animals [11]. Experimental pigs used in advanced surgical procedures will need a wider range of analgesics than those marketed for production animals. Currently, it seems that both choices of drugs and dosages are commonly extrapolated from humans to laboratory pigs [5]. Even though the need for post-operative analgesia should be considered at least the same as for humans undergoing surgery, the direct translation of medicines and dosages from other species to pigs could result in failure to fulfil analgesia, and thus a breach of the refinement principle. Pain will negatively impact animal welfare and the physiological and psychological effects of pain can bias experimental data by adversely affecting multiple biological systems [12]. This highlights the importance of appropriate use and scientific reporting of analgesic agents [7].

Opioids are frequently used in both human and veterinary medicine as analgesic agents pre-, intra-, and post-operatively. Buprenorphine and fentanyl are among the most used opioids to relieve post-operative pain in laboratory pigs [5]. Fentanyl transdermal patches and transdermal solutions have both been studied as post-operative analgesic agents in experimental pigs as these formulations negate the need for repeated handling and injections [13,14,15,16]. Fentanyl is efficient for relieving moderate to severe pain in many mammalian species [17,18], but evidence for its efficacy in pigs is rather limited, with a wide variation in plasma concentration and effects on physiological parameters, activity level and behavior being reported [13,14].

Evaluation of post-operative pain in pigs is largely based on the assessment of behavior [5,19], but because opioids can affect both behavior and activity level this may interfere with pain evaluation. Side effects secondary to opioids are well documented in horses [20,21,22], cats [23], and a recent study in pigs also demonstrates anxiety-like behavior and increased movement after butorphanol injections [24].

The mechanisms behind the side effects altering behavior and locomotion seem to differ between different opioids. When it comes to fentanyl, some of the observed side effects may be mediated by serotonin, as fentanyl has been shown to increase serotonin efflux in the dorsal raphe nucleus [25], and been coupled to serotonin syndrome-like behavior in rats [26]. In humans, serotonin syndrome has been reported when fentanyl is combined with other serotonergic drugs [27,28]. Serotonin syndrome is an iatrogenic drug-induced toxidrome, which is characterized by neuromuscular and autonomic hyperactivity [28,29]. Some of the locomotory and behavioral changes seen as a response to opioid administration in pigs could therefore be explained by an opioid-induced increase in serotonin levels. The fact that serotonin plays a role in motor control [30] strengthens this hypothesis. The effect of serotonin on motor control depends both on serotonin levels and the receptor subtype activation: 5HT_2A_ receptors are present in the postsynaptic compartment in the spinal cord, and activation of this receptor has been coupled to increased excitability of motoneurons [31].

Increased knowledge about the effects opioids have on behavior and locomotion in pigs is essential for assessing post-operative pain and analgesic requirements. The primary aim of this study was therefore to test the effect fentanyl has on locomotion and behavior in freely moving pigs. As a secondary aim we tested whether any effects could be linked to 5HT_2A_ receptor activation, by using ketanserin, a 5HT_2A_ receptor antagonist. We predicted that injection of fentanyl would increase locomotor behavior in freely moving pigs compared to a control group, and that ketanserin would reduce the effect of fentanyl on locomotion.

## 2. Materials and Methods

The study was approved by the Norwegian Food Safety Authority (FOTS ID: 29067). The experiment was planned according to the ARRIVE guidelines.

### 2.1. Animals and Housing

An a priori power estimation was performed using power explorer for two independent sample means (JMP Pro 16.0.0, SAS Institute Inc., Cary, NC, USA), with effect size and standard deviation retrieved from a pilot study. To detect a minimum of 33% difference in movement from baseline after two injections of fentanyl, have a power of 90% and an alpha of 0.05, a control group of four pigs, and a fentanyl group with 10 pigs, was necessary.

A group of 14 mixed-breed pigs (25% Norwegian Landrace, 25% Yorkshire and 50% Duroc), were housed in pairs of two at the research animal facility at the Norwegian University of Life Sciences. There were five surgically castrated males and nine intact females from two different litters included. The pigs originated from the Livestock Production Research Centre of the Norwegian University of Life Science and were 59 ± 1.2 days old (mean ± standard deviation) with a body weight of 21.1 ± 2.4 kg (mean ± standard deviation) on the day of the experiment. The experiment was carried out in two blocks with six and eight pigs per block from March to May 2022. The pigs were allowed a one-week acclimatization period before the experiment.

Within the animal housing rooms pens were constructed either as two single pens of 2.86 m^2^, or as a single 5.78 m^2^ pen with a moveable door which made it possible to divide the larger unit into two units of 2.86 m^2^. The pigs were housed in pairs of two in either the larger 5.78 m^2^ pen, or two pigs in the smaller 2.86 m^2^ pen before and between experiments. During the experiment, each pig was housed singly in the 2.86 m^2^ pen. Peat (Naturtorv, Floralux, Nittedal, Norway) was used as bedding. The pigs had free access to hay and water at all times and were fed a commercial grower diet (Format Vekst 110, Felleskjøpet, Lillestrøm, Norway) twice daily. Humidity and temperature were controlled and kept at 40–50% and 18–20 °C, respectively. A 12:12 h light: dark cycle, with a 30-min transition period, was used. Pigs were checked a minimum of twice daily for the whole study period by a trained animal technician.

### 2.2. Socialization, Enrichment and Training

Pigs were provided with toys (plastic balls, plastic dog toy, paper, newsletters), treats and rooting material as environmental enrichments. Their rooting material consisted of peat and straw which covered the floor to a depth of 5–10 cm. During the acclimatization period each pig was individually trained to get used to human contact and to follow a target stick using positive reinforcement. This allowed later stress-free handling and made procedures such as injecting through the ear vein catheter easier to perform. The training program was divided into four steps, and adjusted according to a previous training program performed in pigs used for renal transplantation studies [32]. The program is illustrated in Table 1. Step one lasted until the pigs were familiar with the presence of the trainer in the pen area and did not attempt to avoid or flee from the human. Step two lasted until the pigs were familiar with treats, then a target stick was introduced in step three. In step four the target stick was used to habituate the pigs to procedures using positive reinforcement. Two or three persons were involved in the training program, and it lasted for approximately one hour per day, five days a week for each pig.

### 2.3. Study Design

The study was designed as a blinded prospective, balanced three-group study. Upon arrival at the animal research facility, pigs were housed in pairs of two and received a number between three and 16. A block randomization into three groups (fentanyl–ketanserin (FEN+KET), fentanyl–saline (FEN+SAL) or saline–saline (SAL+SAL)) was performed, to ensure that pigs housed in the same room received either saline or fentanyl for the two first injections.

### 2.4. Drugs and Chemicals

Ketanserin tartrate (Tocris bioscience, batch no.: 4B/264887) was prepared according to manufacturers’ instructions with sterile water (sterile water, Fresenius Kabi, Verona, Italy). Solubility was obtained by gentle warming (40–60 °C) and rapid stirring. After being dissolved in sterile water, the ketanserin solution was diluted with hypertonic saline (hypertonic saline 7.2%, Covetrus, Portland, NE, USA) to obtain an isotonic solution of ketanserin, 3 mg/mL. Aliquots of 10 mL were prepared and stored in tightly sealed vials (Cellestar tubes, Greiner bio-one) at −20 °C for a maximum of 1 month. Thawing was achieved with gentle warming (40–60 °C) on the morning of the experimental day.

Four injections were prepared for each pig on the experimental day. The allocation of the different drugs and chemicals between experimental groups and phases is demonstrated in Table 2. Injections 1 and 2 were stored at room temperature, while injections 3 and 4 were kept in a warm water bath (40–60 °C) until used. The persons responsible for preparing the injections were not involved in the experimental procedure.

Fentanyl (fentanyl 50 µg /mL, Hameln) was prepared at a dosage of 5 µg/kg for the first injection, and 10 µg/kg for the second injection. The 5 µg/kg was injected to reflect a clinical dosage, with a target plasma level between 0.5–2 ng/mL based on a previous study using fentanyl in pigs [33]. This concentration of plasma fentanyl is a range associated with analgesia in humans [15,34]. The second dose of 10 µg/kg was included based on a pilot experiment in order to make sure that behavioral changes would be induced in all pigs and thus allow a full evaluation of a possible reversal potential by ketanserin. The dose of ketanserin was chosen based on previous studies [35,36,37] and the pilot experiment.

Saline (natriumklorid 9 mg/mL, Fresenius Kabi) was prepared in an equivalent volume as fentanyl (5 µg/kg or 10 µg/kg) for injections 1 and 2, and ketanserin (1 mg/kg) for injections 3 and 4.

Injection 4 was included to ensure that all pigs under fentanyl influence received ketanserin as an antagonist.

### 2.5. Anesthesia and Central Venous Catheter Placement

The day before the experiment pigs were examined clinically (general impression, cardiac and pulmonary auscultation) and included if they met the preset inclusion criteria. The pig was transported into the operating room in an animal transport trolley. Anesthesia was induced and maintained with isoflurane (IsoFlo vet 100%, Zoetis) vaporized in 100% oxygen administered by a facemask (Midmark, Versailles, OH, USA). An anesthetic monitor (GE Healthcare Monitor B650; GE Healthcare, Helsinki, Finland) was used to display peripheral oxygen saturation (SpO_2_) and pulse frequencey by a pulse oximeter probe placed on the dew claw of a front leg.

The ear was aseptically prepared with antiseptic soap (Hibiscrub, Mölnlycke, Quetigny, France) and chlorhexidine in ethanol (klorhexidinsprit 5 mg/mL, Fresenius Kabi). Then a central venous catheter (Careflow 2.5 or 3 F, 200 mm, Merit Medical, Yishun, Singapore) was placed in the lateral ear vein using a sterile Seldinger technique and sutured to the skin using a monofilament non-absorbable suture (Ethilon 2-0, San Lorenzo, Ethicon, Puerto Rico) before being covered by adhesive bandage material (Tensoplast, Essity, Pinetown, South Africa) and an outer dressing(Animal polster, Snögg, Vennesla, Norway).

After catheter placement, the pigs were transported back to the home pen. The pigs were housed separately in single pens (2.86 m^2^) for 24 h but had auditory and olfactory contact with each other.

### 2.6. Experimental Procedure

Each pair of pigs went through the experimental procedure at the same time. Experiments were carried out between 9 am and 2 pm. Food and toys were removed 1 h before the start of the experiment. Both pigs were examined clinically and only included if they met the inclusion criteria. The catheter was inspected and flushed with 5 mL heparinized saline (4 IE/mL Heparin, Leo) to ensure a patent central venous access. The test area is illustrated in Figure 2 and was the same area as the home pen separated by a movable wall. Observation and data collection was from the front side of the pens (observer area) and were carried out by the same persons for all pigs (ND, JN).

Just before the start of the baseline recording, the experimental personnel went into the pens of the two experimental pigs, touched their ears, spoke to them, and gave them a reward. Then baseline recording started, and all further observation was made from the observation area. The baseline video recording (Axis m1124-e network camera, Noldus, The Netherlands) lasted for 20 min, and the following recordings, one after each injection, lasted for 10 min. During each recording, the respiratory frequency was counted manually after one and five minutes, and vocalization was counted with a hand counter after three and eight minutes. Injections one to four were administered through the central venous catheter. This was performed without any restraint by the experimental personnel. After the experiment, the central venous catheter was removed, and the pigs were reunited. The experimental timeline from arrival until the end of the experiment is illustrated in Figure 1.

### 2.7. Data Collection and Processing

Videos were analyzed in Ethovision XT (Version: 9.0.726, Noldus Information Technology, Netherlands). An arena for each pig was created, with a calibrated area of 217 × 132 cm (Figure 2). Detection settings for each pig were individually adjusted. The detection method was dynamic subtraction, with the subject brighter than the background. Contrast and subject size were fine-tuned for each pig to optimize tracking. The sample rate was set to 10 per s.

Locomotion was quantified as distance moved and tracked by center-point detection. The maximum proportion of samples with non-detection was 0.1%. Distance moved was calculated as the total distance walked (cm) over 10 min. For the 20-min-long baseline recording, distance moved (cm) was calculated for the first and last 10 min and averaged for further analysis.

Other behaviors were scored manually in Ethovision. The ethogram, and the variables extracted for statistical analysis, are described in Table 3. The same researcher (ND) scored all videos and was blinded with regards to the treatment of the pigs.

### 2.8. Data Analysis

Data were analyzed using JMP Pro 16.0.0 (SAS Institute Inc., Cary, NC, USA). A mixed model was used to test whether there was a significant difference in distance moved between pigs receiving fentanyl and saline. This was the first aim of the experiment. Each pig was included as a random factor, nested in treatment. Treatment and time, and the treatment by time interaction were included as fixed factors. To obtain normality of residuals and homogeneity of variance, the distance moved had to be square root transformed. Three post-tests were carried out. Correction for multiple testing was performed with the Bonferroni method, which yielded a new critical *p*-value of 0.017. Our second aim was to investigate if ketaserin influenced the distance moved by pigs that had received fentanyl. This analysis was carried out on a subset of the pigs: only those that had received fentanyl as the two first injections (n = 10). The difference (Δ) in distance moved before and after injection 3 with either ketanserin or saline was calculated and compared with a Welch’s unequal variances *t*-test. A graphical illustration of the different statistical models used in the different subset of pigs is demonstrated in Figure 3.

The results for the other behaviors in the ethogram, as well as respiratory frequency and vocalization, are presented by descriptive statistics. Since normal distribution could not be assumed the data are presented as median and range. The behaviors described for each pig are summarized for each experimental timeslot (baseline, injection 1–3).

## 3. Results

Distance moved in response to the two injections of fentanyl (5 µg/kg and 10 µg/kg) increased compared to baseline, and compared to saline receiving pigs at the second injection. This is shown in Figure 4. The injection of ketanserin did not significantly reduce the distance moved in fentanyl treated pigs (Figure 5).

The gait/posture pattern changed from a normal pattern to a stiff gait/posture pattern in response to fentanyl administration. Injection of ketanserin induced a transient ataxic pattern which lasted for 1–5 min before pigs returned to the normal gait/posture pattern. The transition between different gait/posture patterns in response to treatment is demonstrated in Figure 6.

Pigs receiving fentanyl showed a freeze behavior which was reversed by injection of ketanserin, as illustrated in Figure 7. All pigs also showed repetitive stereotypic patterns of behaviors after fentanyl injection. These repetitive behaviors consisted of frequent drinking/spilling water, circling, backward locomotion, excessive rooting behavior or jumping. Each pig exhibited at least one of these seemingly non-goal-directed behaviors. Which behavior(s) the individual pigs displayed is illustrated in Figure 8 and summarized in Appendix A.

Behaviors which remained stable among all pigs at baseline, and throughout the study period for the saline–saline group were play behavior, resting (sternal and lateral recumbency) and sitting. Duration of play behavior and sitting and of sternal and lateral recumbency are displayed in Figure 9. Play behavior, resting and sitting dropped markedly after fentanyl injection. Resting partly returned after ketanserin administration but play behavior did not. Respiratory frequency increased in response to fentanyl (Figure 10). Vocalization remained stable within the groups and the median (range) for the vocalization frequency for all experimental phases was 4 (0–34) for fentanyl and 4 (0–18) for saline treated pigs.

The median (range) duration or frequency of all behaviors for each treatment group and experimental phase are provided in Appendix A. Data from injection 4 were not included in the statistical analysis due to the time difference, which made statistical comparison infeasible, but is included in Appendix A.

## 4. Discussion

This study demonstrates that fentanyl significantly increased locomotor activity in freely moving, non-painful pigs. Distance moved increased from baseline after both the first and second injections of fentanyl. Distance moved after the second dose of fentanyl was significantly different from the saline–saline group which served as a control. These results are consistent with what has been described in response to opioid administration in horses [38], where this adverse reaction partly accounts for the limited clinical use of this class of analgesic agents. Increased locomotor activity secondary to fentanyl is also reported in cats, both after intravenous injections [39], and after transdermal application [23]. In foals, a positive dose–response relationship with regards to locomotor activity and fentanyl administration has been demonstrated [40]. In this current study, pigs received two different dosages of fentanyl to reflect both a clinically applicable dose (5 µg/kg) and one that could exaggerate any effects on locomotion and behavior (10 µg/kg). As both injections increased locomotor activity, this highlights the importance of correct interpretation of activity level when fentanyl is used in pigs, as this analgesic agent per se can influence activity.

Distance moved was not significantly reduced in response to ketanserin in the fentanyl-treated pigs. Previous studies have reported increased serotonin efflux after fentanyl administration in rodents [25], and in vitro studies have shown that fentanyl has a direct receptor affinity for the 5HT_1A_ (K_i_ 2.1 µM) and 5HT_2A_ (K_i_ 1.3 µM) [41] receptors. Serotonin acts as a neuromodulator for ionotropic input at motoneurons in the spinal cord [42]. 5HT_2_ receptor activation has been demonstrated to be involved in the excitability of spinal motoneurons in turtle spinal cord preparation [31]. Thus if fentanyl, directly or indirectly, influences the intrasynaptic levels of serotonin, it could also influence the excitability of the spinal motoneurons [43]. Ketanserin is a quinazoline derivative and acts as a selective 5HT_2A_ receptor antagonist [44]. Even though ketanserin administration did not reverse the increased distance moved in fentanyl-treated pigs in the current study, this does not rule out a role of serotonin in fentanyl-induced locomotor activity. The role of the serotonergic system in motor control is extensively studied [30,45], and in addition to the 5HT_2_ receptor, the 5HT_7_ has been coupled to excitatory locomotory drive in a neonatal mouse model [46]. Based on this, targeting the 5HT_2A_ receptor selectively might only partially counteract fentanyl-induced locomotor effects.

Besides affecting distance moved, fentanyl induced a transition from a normal gait/posture pattern to a stiff gait/posture in this study, along with a reduction in resting behavior. The stiff gait and posture pattern was characterized by a small stepping gait, with stiffly extended limbs. The adapted gait/posture pattern induced by fentanyl gave the impression of pigs being in a restless and excited state. Similar findings are described in pigs receiving butorphanol, where restlessness and distress behaviors were reported [24]. As a response to injection 3, pigs in the fentanyl–ketanserin group adapted a transient ataxic gait/posture which transitioned to a normal gait/posture within 1–5 min. Pigs in the fentanyl–saline group which received saline as injection 3 did not present with the transient ataxia but had a reduction in the duration of the stiff gait/posture pattern. The transient ataxia seen in pigs receiving ketanserin can possibly be explained by the blood pressure reducing capacity of the antagonist which is thought to be mediated primarily by antagonist action on the 5HT_2A_ receptor on arteriolar smooth muscle cells, and a possible weak α_1_-adrenergic receptor blocking effect [44,47,48]. Considering the short duration of the visible ataxia in this study, a transient drop in blood pressure seems like a plausible explanation. A similar transient drop in blood pressure secondary to ketanserin injection was seen in a study conducted in pigs under anesthesia ([49]., manuscript in preparation). The stiff gait/posture pattern was slightly reduced in pigs receiving saline (fentanyl–saline group), which could be due to reduction in plasma fentanyl levels over time. Ketanserin seems to have a reversal potential on the stiff gait/posture in pigs receiving fentanyl, which suggest a role of serotonin in this behavior. Even though an effect of transient blood pressure reduction in this immediate reversal cannot be ruled out.

Freeze behavior, as well as high-pitched vocalization and escape attempts have been linked to negative emotions in pigs [50]. Pigs receiving fentanyl adopted a freeze behavior which was reversed by injection with ketanserin. The freeze response resembled the startle and freeze reflex, which is a protective response in many species and elicited by a potentially threatening stimulus. A previous study demonstrates that pigs display this behavior in response to a novel sound stimulus [51]. In the current study no novel stimuli were provided, but pigs also exhibited behaviors such as frequent jumping towards the wall, which by the observers were thought to resemble escape behavior as it looked like the pig was trying to get out of the pen (video in Appendix A). Similar jumping patterns, along with restlessness, have been demonstrated in pigs receiving butorphanol, and the authors suggested anxiety to be the underlying emotion [24]. Serotonin can influence neuromuscular and autonomic activity as well as mental status [27]. Rodent studies have demonstrated that fentanyl can be associated with serotonin syndrome-like behavior [26]. And the role of the 5HT_2A_ receptor in opioid induced behavior is further supported by a rodent study where alfentanil induced an exaggerated startle response and muscle rigidity, whereas ketanserin pre-treatment resulted in an absent startle response and flaccid movements [36]. Freeze behavior was also reversed by ketanserin in this study, which could suggest a role of the 5HT_2A_ receptor in this behavioral trait. Both the freeze and jumping behavior, along with the stiff gait/posture, gave the impression of pigs being in an excited, hypervigilant state under fentanyl influence.

Central nervous system (CNS) excitation accompanied by increased respiratory frequency have been described in both pigs [24] and horses [52] after administration of butorphanol and buprenorphine. An increased respiratory frequency was observed in the current experiment, with panting in five out of ten pigs in response to fentanyl. Panting has also been described in pigs after application of fentanyl transdermal solution [16], and secondary to butorphanol injections where it was coupled to increased body temperature [24]. Opioids can influence temperature, with increased body temperature in cats [53], and decreased body temperature in dogs [54]. A role of opioids in altering the thermoregulatory set point may be the underlying mechanism of panting [55]. On the other hand, increased motor activity can also influence body temperature and CO_2_ production, with increased respiratory frequency as a compensatory mechanism.

Contrary to the negative emotional expression in pigs receiving fentanyl, pigs in the saline–saline group showed resting and play behavior throughout the study. Play is coupled to positive emotions [50], and has been used as an indicator of good animal welfare [56]. Pigs in the fentanyl–ketanserin and fentanyl–saline groups only displayed play behavior at baseline, and ketanserin did not rescue this positive behavior. Play behavior is complex, and animals might take some time to regain motivation to play after an adverse experience, which the fentanyl–injection seemed to be.

How animals are affected by opioids does not only seem to vary between species, but also between individuals within the same dose range [40,57]. This was also the case in the current study, where individual pigs adapted different repetitive, seemingly non-goal-directed, behavioral patterns in response to fentanyl, which persisted throughout the study. Repetitive snout-contact behavior has earlier been seen in pigs in response to butorphanol [24], which resembles our observation of excessive rooting behavior in some of the pigs. Five out of the ten fentanyl treated pigs showed persistent circling. This is also demonstrated in horses following morphine administration, where seven out of ten horses showed persistent circling after intravenous injection of 0.05–0.1 mg/kg morphine [58]. Different hypotheses have been put forward to explain both locomotor and behavioral side effects in different species secondary to opioids, including proposed roles for dopamine neurotransmission [57], and opioid metabolites (MG3) [58] in horses, and a different affinity for opioid receptor subtypes (µ_1_/µ_2_) [16], and the central role of κ-agonist and GABA-interneuron inhibition in dysphoria and anxiety-like behavior [24] in pigs. Ketanserin reduced the frequency of some but not all the repetitive behaviors (Figure 7), and more research is needed to understand the relative importance of the 5HT_2A_ receptor in these behaviors compared to other neurochemical pathways.

The aim of this study was to demonstrate potential effects on locomotion and behavior when fentanyl is administered to pigs, and we therefore conducted the study in non-painful pigs. This must be considered when interpreting and extrapolating our results as the presence of pain is believed to influence the occurrence of side effects [59]. Experimental studies have demonstrated that methadone injection induced dysphoria in non-painful dogs [60], while dogs subjected to surgery in another study did not present dysphoria when methadone was administered post-operatively [61]. This is also described in pigs receiving fentanyl where they reported adverse reactions in the preliminary step without surgery, but not in the post-operative phase [16]. Further research on the effect of fentanyl in pigs is necessary to test whether the side effects are fully present or reduced in painful conditions. As the target variable was distance moved after fentanyl and ketanserin administration, a decision was made not to take plasma samples to measure fentanyl concentration, as this would have interfered with the pigs’ behavior.

## 5. Conclusions

In conclusion, fentanyl injections induced changes in both locomotor activity and behavior at both dosages administered. The reversal potential by ketanserin on some of the induced behavioral changes suggests that the 5HT_2A_ receptor plays a role in some of the fentanyl-induced side effects. As a clinical implication, if behavior and activity level are used to evaluate analgesia in pigs, caution should be made with regards to how they may be influenced by fentanyl, because importantly, providing analgesic drugs does not directly translate to providing analgesia.

## 6. Patents

A patent application has been disclosed by the authors in light of the current research work.

## Figures and Tables

**Figure 1 animals-13-01671-f001:**
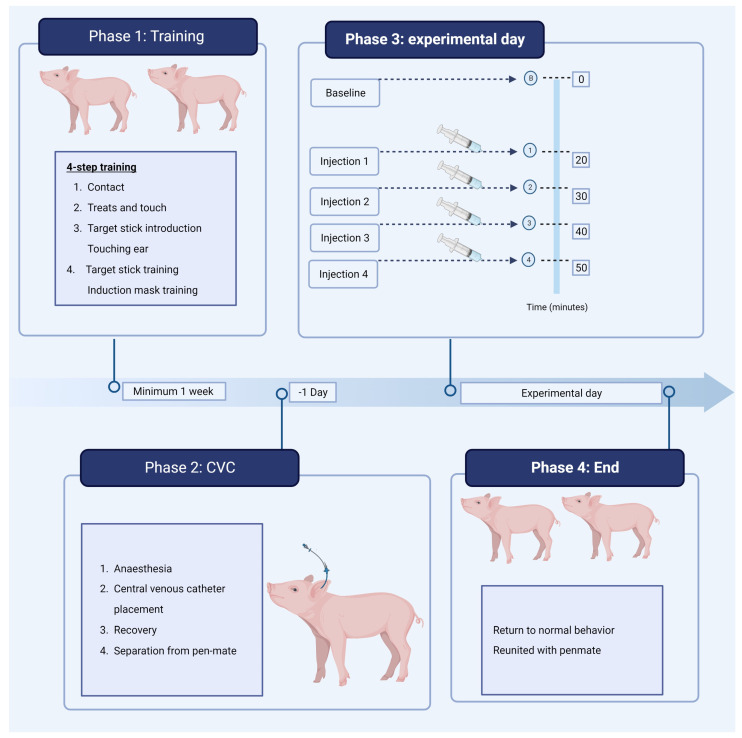
Experimental timeline from arrival until the end of the experiment. CVC: central venous catheter positioned in lateral ear vein. Created with BioRender.com.

**Figure 2 animals-13-01671-f002:**
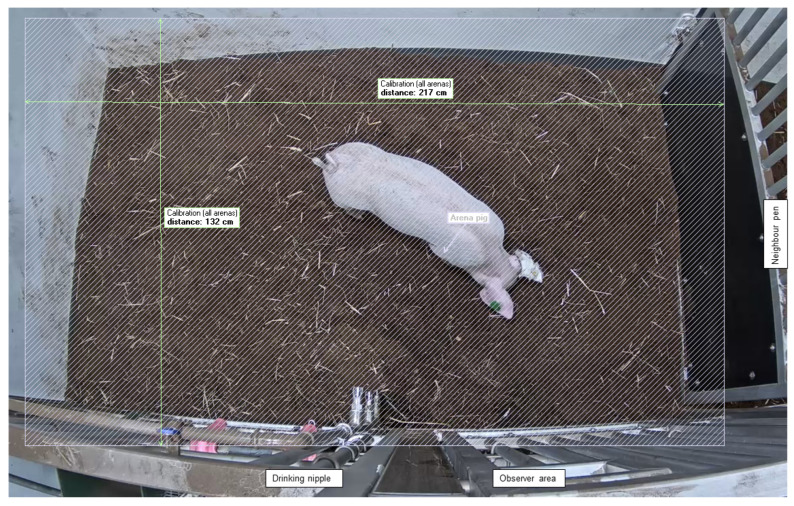
Test pen area. Image retrieved from Ethovision XT (Version: 9.0.726, Noldus Information Technology, Wageningen, The Netherlands).

**Figure 3 animals-13-01671-f003:**
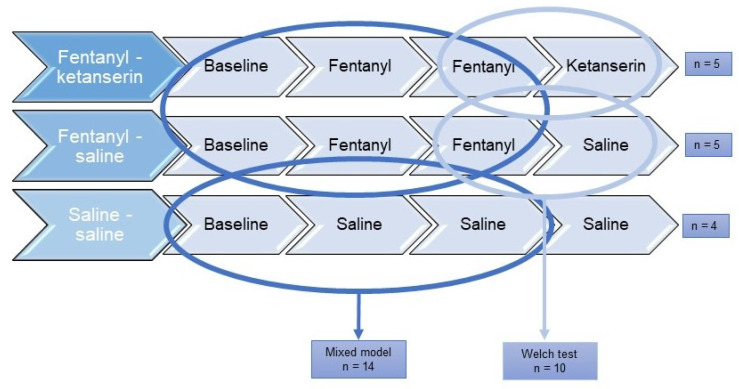
Graphical illustration of treatment groups used for statistical analysis. The mixed model was used to test the effect of fentanyl on locomotion, whereas the Welch’s *t*-test was used to test the effect of ketanserin in pigs administered fentanyl. n = number of experimental pigs.

**Figure 4 animals-13-01671-f004:**
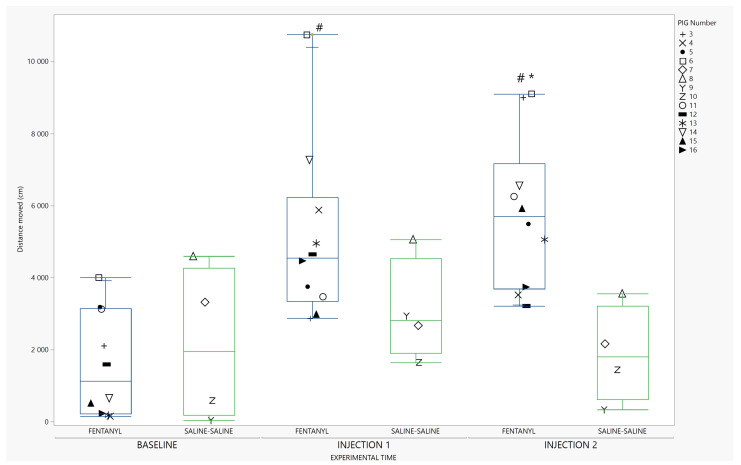
Distance moved in response to fentanyl. Distance moved (cm) during baseline and after injection 1 and 2 for pigs receiving fentanyl (blue), or saline (green) is illustrated with a box-and-whisker plot with interquartile range, and median illustrated within the box. Upper and lower range by whiskers as vertical lines extending from the box. Distance moved is summarized for 10 min within each timeslot. Data from pigs in the fentanyl–ketanserin (n = 5) and fentanyl–saline group (n = 5) are reported together as one group named Fentanyl (n = 10) Data from each pig are presented with a marker within the box. The marker assigned to each individual pig (coded 3–16) is shown on the right panel. * Significantly different from saline–saline group (*p =* 0.0027 for pairwise comparison; F _(treatment*time) (2,24)_ = 5.68; *p* = 0.0095). # Significantly different from baseline (*p* = <0.001 for pairwise comparison for injection 1 and injection 2). (Single column fitted figure).

**Figure 5 animals-13-01671-f005:**
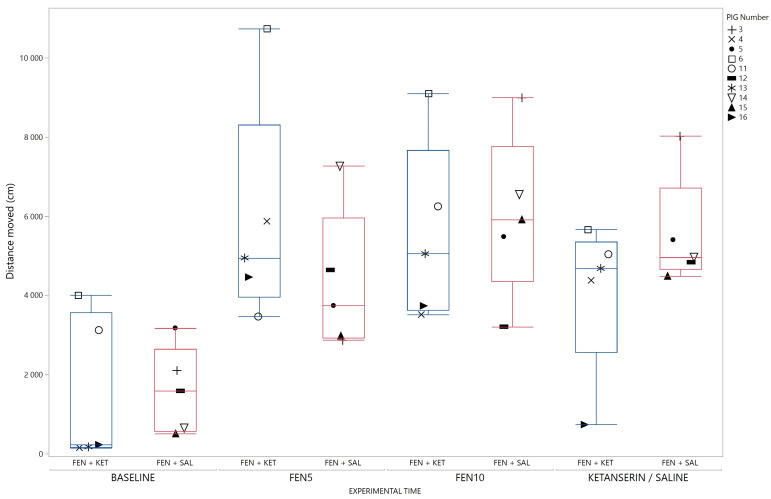
Distance moved in response to ketanserin in fentanyl treated pigs (n = 10). The figure illustrates distance moved (cm) from baseline, and in response to injection of 5 µg/kg fentanyl (FEN5), 10 µg/kg fentanyl (FEN10) and either ketanserin (1 mg/kg) for the fentanyl–ketanserin (FEN + KET) pigs (blue), or saline for the fentanyl–saline (FEN + SAL) pigs (red). Results are illustrated with a box-and-whisker plot with interquartile range, and median illustrated within the box. Upper and lower range by whiskers as vertical lines extending from the box for each group and experimental phase. Data from each pig are presented with a marker within the box. The marker assigned to each individual pig (coded 3–16) is shown on the right panel. The difference (Δ) between injection 2 (FEN10) and 3 (KETANSERIN/SALINE) for the fentanyl–ketanserin and fentanyl–saline group was calculated and compared with a Welch’s unequal variances *t*-test. Reduction in distance moved in response to ketanserin or saline was not statistically significant (F (treatment) (1, 7.3) = 0.9; *p* =0.37). (Single column fitted figure).

**Figure 6 animals-13-01671-f006:**
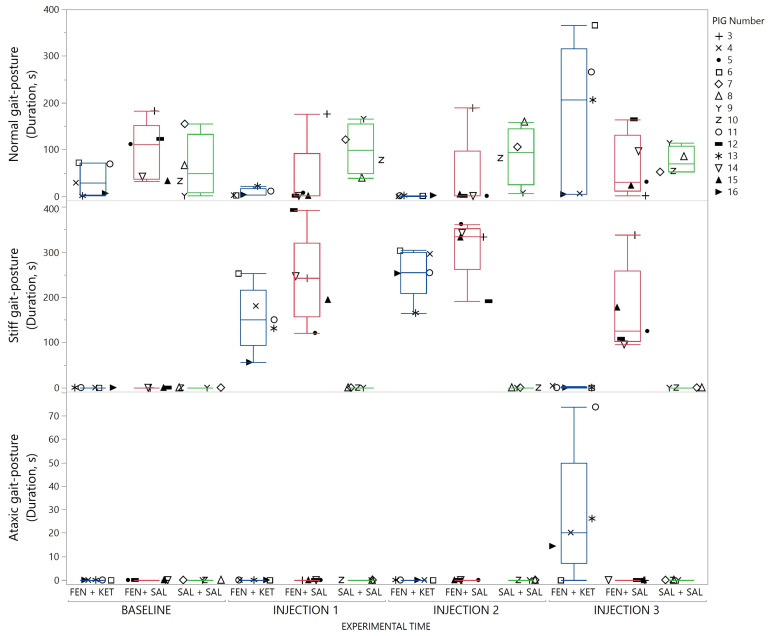
Gait/posture patterns displayed by the pigs in response to treatment and time illustrated with a box-and-whisker plot (with interquartile range, and median illustrated within the box. Upper and lower range by whiskers as vertical lines extending from the box). With duration (seconds (s)) of the different patterns. Data for each timeslot, baseline, injection, 1, 2 and 3, are summarized as cumulative duration for 10 min. Pigs in the saline–saline (SAL+SAL) group (green) received saline for all injections. Pigs in the fentanyl–ketanserin (FEN+KET) group (blue) and fentanyl–saline (FEN+SAL) group(red) received fentanyl for injection 1 and 2, and then ketanserin or saline respectively for injection 3. Data from each pig are presented with a marker within the box. The marker assigned to each individual pig is shown on the right panel (1.5 column fitted figure).

**Figure 7 animals-13-01671-f007:**
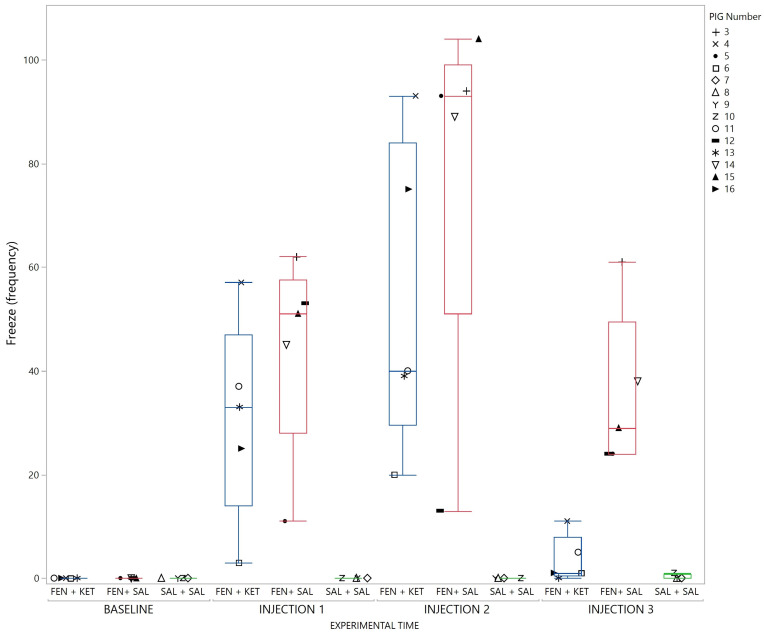
Freeze behavior illustrated with a box-and-whisker plot with interquartile range, and median illustrated within the box. Upper and lower range by whiskers as vertical lines extending from the box. Frequency within the different groups is summarized for 10 min within each timeslot; baseline, injection 1, 2 and 3. Treatment groups; fentanyl–ketanserin (FEN + KET), fentanyl–saline (FEN + SAL) and saline–saline (SAL + SAL). Data from each pig are presented with a marker within the box. The marker assigned to each pig is shown on the right panel (Single column fitted figure).

**Figure 8 animals-13-01671-f008:**
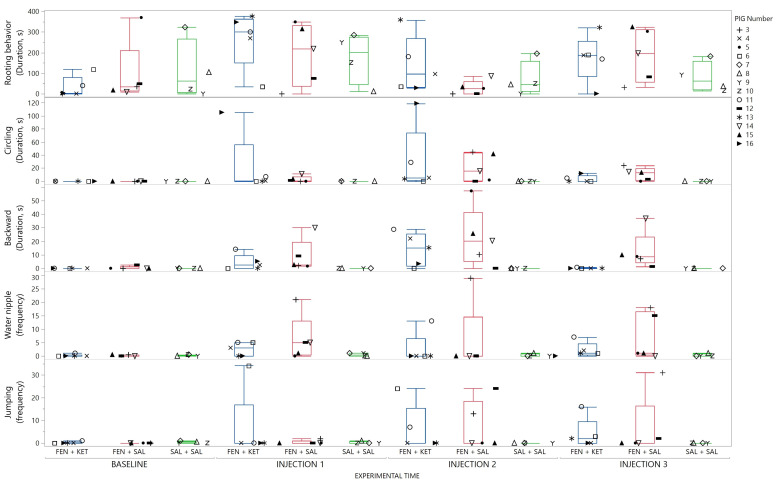
Repetitive behaviors displayed by each pig (3–16) at each timepoint (baseline, injection 1–3). Repetitive behaviors were rooting behavior, circling, backward locomotion, frequency of being in contact with the water nipple and jumping. The sum of duration or frequency for each behavior at each timepoint is displayed for each pig with a marker within the box- and whisker plot. The box- and whisker plot summarizes the median and IQR of each group at each experimental time-point. The marker assigned to each pig is shown on the right panel. (Double column fitted figure).

**Figure 9 animals-13-01671-f009:**
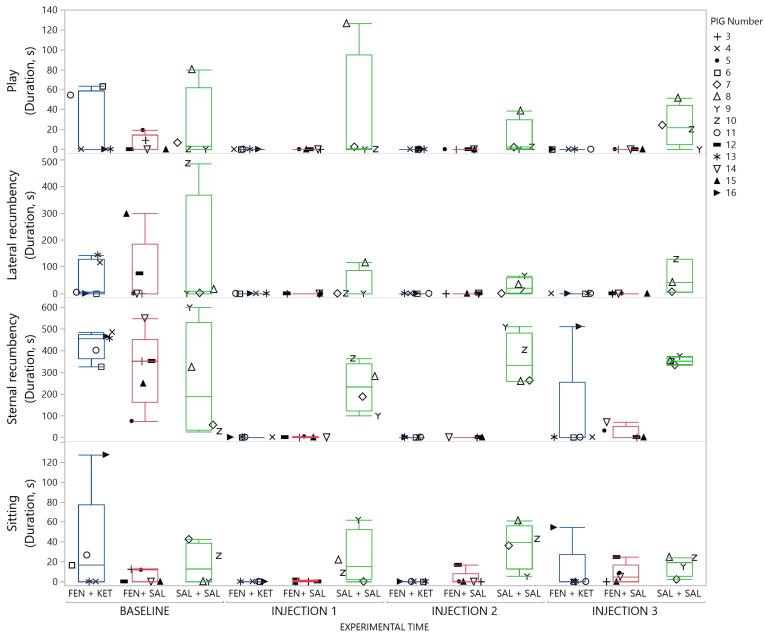
Play and resting behavior. Distribution of play, lateral and sternal recumbency and sitting behavior demonstrated with a box-and-whisker plot, with interquartile range, and median illustrated within the box. Upper and lower range by whiskers as vertical lines extending from the box. Duration (s) performing these behaviors in response to treatment and time is illustrated. Data for each timeslot (baseline, injection 1, 2 and 3) in each group (fentanyl–ketanserin (blue), fentanyl–saline (red) and saline–saline (green)) are summarized for 10 min. Data from each pig are presented with a marker within the box and whisker plot. The marker assigned to each pig is shown in the right panel (1.5 column fitted figure).

**Figure 10 animals-13-01671-f010:**
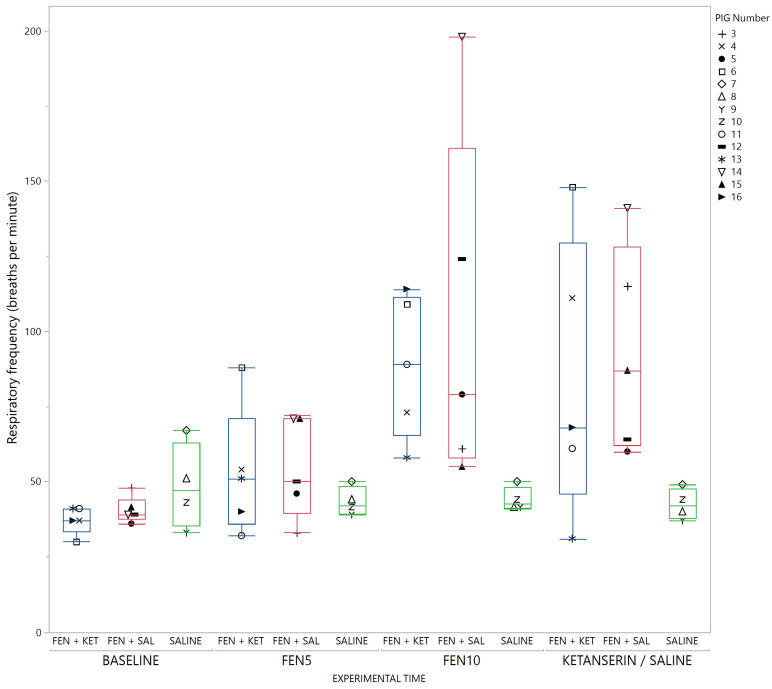
Respiratory frequency, given as breaths per minute in response to treatment and time. Data for each timeslot (baseline, injection, 1, 2 and 3) in each group (fentanyl–ketanserin (blue), fentanyl–saline (red) and saline–saline (green)). Data are demonstrated with a box-and-whisker plot with interquartile range, and median illustrated within the box. Upper and lower range by whiskers as vertical lines extending from the box. Data from each pig are presented with a marker within the box. The marker assigned to each pig is shown on the right panel (Single column fitted figure).

**Table 1 animals-13-01671-t001:** Four-step training program.

Steps	Trainer Intervention
Step 1	Trainer enters the pen area. Pigs are allowed to adapt to human presence. Trainer sits in the pen without interacting with the pigs.
Step 2	Introduction of treats (apples, grapes, and Norwegian unleavened bread). Tossing treats in the pen, followed by providing treats by the hand of the trainer. Touching the back, ears and the head of pig while providing treats. Talking to the pigs.
Step 3	Introduction to target stick by positive reinforcement rewarded by treat when touching the target.
Step 4	Training pigs with target stick to stand still, touching their ears, and move within the animal experimental facility and into the transport trolly. Accustom pigs to facemask with positive reinforcement.

**Table 2 animals-13-01671-t002:** Allocation of drugs and chemicals for each experimental group and phase. n = number of experimental animals.

Phase Experimental Group	Baseline	Injection 1	Injection 2	Injection 3	Injection 4
Fentanyl-ketanserin (*n* = 5)	-	Fentanyl 5 µg/kg	Fentanyl 10 µg/kg	Ketanserin 1 mg/kg	Saline
Fentanyl-saline (*n* = 5)	-	Fentanyl 5 µg/kg	Fentanyl 10 µg/kg	Saline	Ketanserin 1 mg/kg
Saline-saline (*n* = 4)	-	Saline	Saline	Saline	Saline

**Table 3 animals-13-01671-t003:** Ethogram used to score behavior. The ethogram was created based on observations from a pilot study of four pigs.

Behavior	Description	Outcome Variables (per 10 min)
Normal gait-posture	Normal gait and posture. Normal balance and length of steps.	Duration (s)
Stiff gait-posture	Small movements of limbs and body with stiff short steps. Extended legs. “Stepping gait”.	Duration (s)
Ataxic gait-posture	Ataxic gait-posture, with reduced control of hindlegs.	Duration (s)
Lateral recumbency	Lateral recumbency	Duration (s)
Sternal recumbency	Sternal recumbency with head down or up	Duration (s)
Rooting behavior	Exploring peat or floor with snout in-ground	Duration (s)
Play	Playful running, jumping (vertical and horizontal bouncy movements) and rolling	Duration (s)
Backward locomotion	Walking backwards	Duration (s)
Circling	Circling around hindquarters	Duration (s)
Sitting	Sitting with the rear end of the body in contact with the ground.	Duration (s)
Water nipple	Spilling or drinking water	Frequency
Freeze	Fixed body posture with a stiff, extended neck, while staring right ahead	Frequency
Jumping on wall	Attempting to or successfully jumping on wall	Frequency
Undefined	Behaviors not described further such as scratching, rolling in water, defecation/urination	Duration (s)

## Data Availability

Publicly available datasets were analyzed in this study. This data can be found here: NMBU Open Research Data [https://doi.org/10.18710/RENORA].

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
