# Peer review of "High and Hyper: Fentanyl Induces Psychomotor Side-Effects in Healthy Pigs"

_animals, 2023, doi:10.3390/ani13101671_

Round 1

Reviewer 1 Report

Thank you for an interesting study on serotonin in fentanyl-induced psychomotor effects in pigs. See below for some suggestions and questions

Line 22 using changes in behavioral

Line 59 Fugazzola 2022, 1% in a limited field (articular cartilage repair)

Line 78 -79 Rephrase

Line 91 Pavlovsky et al,  Do you mean buthorphanol?

Line 96 Behavioral effects

Line 107 have or effect has

Line 136-137  Before experiment seven pig we housed in pairs (?)  2 and 2 and 3? In one pen?
Seven pigs were hold together in the large pen and seven pigs were hold in groups of 3 and 4 in the small pens? This needs to be clarified.

Line 143  Checked by a veterinarian?

Line 153  Master thesis, Replace with peer reviewed article https://doi.org/10.1177/0023677219879169

Line 165 How many trainers were involved, time per day (approx)?

Line 194 replace were with was, or fentanyl doses were

Line 198 Do you mean concentration in both serum and plasma or is it plasma concentration of fentanyl?

Line 246 What camera is used?

Line 251 Is the experimental personnel same persons as experimenters? Using the same designation will be easier for the reader.

Figure 1  Suggestion to have time on the timeline (days e.g.)

Line 347 that did not seem to serve….has same definition as: repetitive stereotypic patterns of…
one of these is enough

Line 425-426 …have been extensively studied (

Minor editing of English language required

Author Response

Dear reviewer, 

Thank you for a very nice review. Please see the attachment in the box. 

Reviewer 2 Report

Digranes et al. present an assessment of the behavioral effects of intravenous fentanyl in laboratory pigs. Their findings highlight the potential for potent opioids to cause inappropriate locomotor behavior in this species. Generally the manuscript is well written and the findings significant as the behaviors described may interfere with pain assessment. The authors should also be congratulated on the detailed ethographic approach and unbiased quantification of locomotor behavior used in the study. The authors are careful in the interpretation of most of their findings, especially in the admission that the apparent side effects may not be a feature of fentanyl use in painful animals. Clearly this aspect warrants further study.

I have a number of minor comments I would like to see addressed prior to publication.

1. Section 2.3 is difficult to follow. Table 2 and figure 1 are not referred to in the text at this point and it would be helpful to do so. Additionally, table 1 appears wholly unnecessary as the information is provided in the text around line 150 and is also summarised in figure 1.

2. The presentation of data in the manuscript could be improved and made more transparent. Individual values should be shown in the box and whisker plots rather than hiding the data. As there is a repeated measures/time element to most charts, points from the same animal could also be linked with faint lines to show pairing and give clearer insights into individual animal trajectories. This is accounted for in the model by including pig as a random factor, why is it not shown?

3. I find figure 8 confusing. Can this not be presented in the style of the other plots but also with individual data points?

3. I am unclear what is compared to what when the Welch's test is used? Is it the delta from after injection 2 to after injection 3 between groups? The reason I ask, is that from figure 3 it certainly looks as if the antagonist has some effect. Can the exact comparison be clarified? Is it possible to account for pig (i.e paired data before/after ketaserin/saline using a different statistical technique) as this may increase the power to detect a difference. I think this needs to be clarified as there is a rather awkward dichotomy in the results section. Firstly the hypothesis that ketanserin can reduce fentanyl induced locomotor activity is rejected based on statistical testing but then multiple conclusions are drawn using purely descriptive statistics of the ethographic data. 

4. Regarding the effect of Ketanserin. The authors are careful not to overstate their conclusions in the discussion. This is prudent as making any real conclusions about whether this is a spinal seratonergic effect and subsequent direct reversal is virtually impossible given the lack of a saline/saline/ketanserin control group. However, the title still suggest this is a major aim of the study. I would like to see the title focus more on the main aim as the seratonergic results can only be considered speculative.

Author Response

(The authors gave the same response as above.)

Reviewer 3 Report

Very nice and interesting study - a few very small comments

Author Response

(The authors gave the same response as above.)

Reviewer 4 Report

Dear Authors, 

congratulations for the very nice article, I enjoyed the reading. I attach the manuscript with minor suggestions, not obligatory, only if they make sense for you

Author Response

(The authors gave the same response as above.)
